# Transcriptional adaptation in *Caenorhabditis elegans*

**Vahan Serobyan[1]\*, Zacharias Kontarakis[1†], Mohamed A El-Brolosy[1], Jordan M Welker[1], Oleg Tolstenkov[2,3‡], Amr M Saadeldein[1], Nicholas Retzer[1], Alexander Gottschalk[2,3,4], Ann M Wehman[5], Didier YR Stainier[1]\***

[1]Department of Developmental Genetics, Max Planck Institute for Heart and Lung Research, Bad Nauheim, Germany; [2]Institute for Biophysical Chemistry, Goethe University, Frankfurt Am Main, Germany; [3]Cluster of Excellence Frankfurt - Macromolecular Complexes (CEF-MC), Goethe University, Frankfurt Am Main, Germany; [4]Buchmann Institute for Molecular Life Sciences (BMLS), Goethe University, Frankfurt Am Main, Germany; [5]Rudolf Virchow Center, University of Würzburg, Würzburg, Germany

**\*For correspondence:**
Vahan.Serobyan@mpi-bn.mpg.de (VS);
Didier.Stainier@mpi-bn.mpg.de (DYRS)

**Present address:** †Genome Engineering and Measurement Lab, ETH Zurich, Functional Genomics Center Zurich of ETH Zurich, University of Zurich, Zurich, Switzerland; ‡Sars International Centre for Marine Molecular Biology, University of Bergen, Bergen, Norway

**Abstract** Transcriptional adaptation is a recently described phenomenon by which a mutation in one gene leads to the transcriptional modulation of related genes, termed adapting genes. At the molecular level, it has been proposed that the mutant mRNA, rather than the loss of protein function, activates this response. While several examples of transcriptional adaptation have been reported in zebrafish embryos and in mouse cell lines, it is not known whether this phenomenon is observed across metazoans. Here we report transcriptional adaptation in *C. elegans*, and find that this process requires factors involved in mutant mRNA decay, as in zebrafish and mouse. We further uncover a requirement for Argonaute proteins and Dicer, factors involved in small RNA maturation and transport into the nucleus. Altogether, these results provide evidence for transcriptional adaptation in *C. elegans*, a powerful model to further investigate underlying molecular mechanisms.

## Introduction

Transcriptional adaptation is the ability of certain mutations in a gene to modulate the expression of related genes, referred to as adapting genes (*El-Brolosy and Stainier, 2017*; *El-Brolosy et al., 2019*; *Ma et al., 2019*). At the molecular level, the mutant mRNA, rather than the loss of protein function, is responsible for this transcriptional modulation (*Rossi et al., 2015*; *El-Brolosy and Stainier, 2017*; *Sztal et al., 2018*; *El-Brolosy et al., 2019*; *Ma et al., 2019*). According to one model (*El-Brolosy et al., 2019*), the mutant mRNA, via its degradation products, modulates the expression of adapting genes via transcriptional regulators including antisense RNAs and histone modifiers. According to another model (*Ma et al., 2019*), the premature termination codon (PTC) containing mutant mRNA interacts with a histone modifier complex leading to transcriptional upregulation of the adapting gene(s). Sequence similarity with the mutant mRNA determines which genes get upregulated during transcriptional adaptation (*El-Brolosy et al., 2019*). In some cases, the upregulated genes share functionality with the mutated gene leading to functional compensation. However, while transcriptional adaptation is often discussed in the context of genetic robustness (*Rossi et al., 2015*; *El-Brolosy et al., 2019*; *Ma et al., 2019*), it does not always lead to functional compensation (*Rossi et al., 2015*). In addition, transcriptome analyses suggest that even genes with limited sequence similarity with the mutant mRNA can be upregulated during transcriptional adaptation (*El-Brolosy et al., 2019*), although clearly more work is required to determine whether the upregulation of these genes is a direct or indirect effect of transcriptional adaptation.

Understanding the mechanisms of transcriptional adaptation will help us better comprehend why for a given gene some mutations cause disease while others do not (*Castel et al., 2018*). However, despite the importance and growing interest in many aspects of genetic compensation, transcriptional adaptation has currently only been investigated in vertebrates. Thus, it remains unclear whether this phenomenon is observed across metazoans. The evolutionary importance of related genes that have compensatory effects has also been discussed in non-vertebrate eukaryotes (*Conant and Wagner, 2004*; *Plata and Vitkup, 2014*). However, it is not known whether these examples of compensation are due to protein feedback loops or transcriptional adaptation.

Only a few factors are known to be involved in the transcriptional adaptation response thus far, and others, including some involved in RNA processing and transport, are likely required. In addition, it is not clear whether the mechanisms of transcriptional adaptation are common or whether each particular case occurs in a different manner, especially at the step leading to transcriptional modulation. Also, as different paralogs or related genes are expressed in distinct tissues and/or at different times (*Laisney et al., 2010*; *Kryuchkova-Mostacci and Robinson-Rechavi, 2015*; *Radomska et al., 2016*; *Pasquier et al., 2017*; *Jojic et al., 2018*), it is currently unclear whether the expression of adapting genes can appear in tissues where, and/or at times when, they are not normally expressed.

In this study, we provide examples of transcriptional adaptation in *Caenorhabditis elegans* and show the ectopic expression of an extrachromosomal reporter in a tissue where it is not normally expressed. In addition, we analyze these transcriptional adaptation models after RNAi-mediated knockdown of different genes involved in RNA metabolism and find that the upregulation of the adapting genes requires factors involved in the maturation and transport into the nucleus of small RNAs (sRNAs).

## Results

### Examples of transcriptional adaptation in *C. elegans*

Actins are essential structural components of eukaryotic cells as they mediate a wide range of cellular processes (*Pollard and Cooper, 1986*). Actin genes are often present in multiple copies in higher eukaryotic genomes and hints of transcriptional adaptation modulating their expression have been reported. For example, mouse embryonic fibroblasts (MEFs) mutant for β-*Actin* (*Actb*) display increased expression of other Actins including α- and γ-Actin (ACTA and ACTG1/2) as measured by Western blots (*Tondeleir et al., 2012*). Similarly, *Actg1* knockout, but not knockdown, in MEFs leads to an increase in *Acta* mRNA levels (*Patrinostro et al., 2017*), and zebrafish *actc1b* mutants exhibit mild muscle defects because of the transcriptional upregulation of *actc1a* (*Sztal et al., 2018*). Furthermore, *Actg1* mutant MEFs and *Actb* mutant mouse embryonic stem cells (mESCs) display increased mRNA levels of *Actg2* and *Actg1*, respectively, and this upregulation is triggered not by the loss of protein function but by mutant mRNA decay (*El-Brolosy et al., 2019*). Thus, we decided to investigate actin genes in *C. elegans* to test for transcriptional adaptation. The *C. elegans* genome contains five actin genes which display high similarity in their DNA and protein sequences (*MacQueen et al., 2005*). We started by analyzing several mutant alleles for *act-1*, *act-2*, *act-3* and *act-5*, and determined mutant transcript levels. We found significantly reduced *act-5* mRNA levels in *act-5(dt2019)* mutants compared to wild type (*Figure 1A*), likely caused by nonsense-mediated decay (NMD) due to a premature termination codon (ptc) in the first exon (*Figure 1A*, *Figure 1—figure supplement 1*). Most mutant alleles of *act-5* cause severe phenotypes including lethality (*Estes et al., 2011*; *MacQueen et al., 2005*), sterility (*Cui et al., 2004*), and paralysis (*Etheridge et al., 2015*). However, the *act-5(dt2019)* allele, hereafter referred to as *act-5(ptc)*, does not exhibit any obvious phenotype (*MacQueen et al., 2005*), an observation we confirmed. We analyzed the mRNA levels of all actin genes in *act-5(ptc)* mutants (*Figure 1A*, *Figure 1—figure supplement 2*), and observed the upregulation of *act-3* mRNA and pre-mRNA (*Figure 1B*, *Figure 1—figure supplement 3*), consistent with a transcriptional adaptation response.

We also examined the *act-5(dt2017)* partial deletion allele, hereafter referred to as *act-5(Δ1)*, (*Figure 1—figure supplement 1*) and found no significant change in *act-5* mRNA levels in homozygous mutants compared to wild type (*Figure 1A*). Notably, *act-3* mRNA levels in *act-5(Δ1)* mutants were not changed compared to wild type (*Figure 1B*). To further test whether *act-3* upregulation in *act-5*

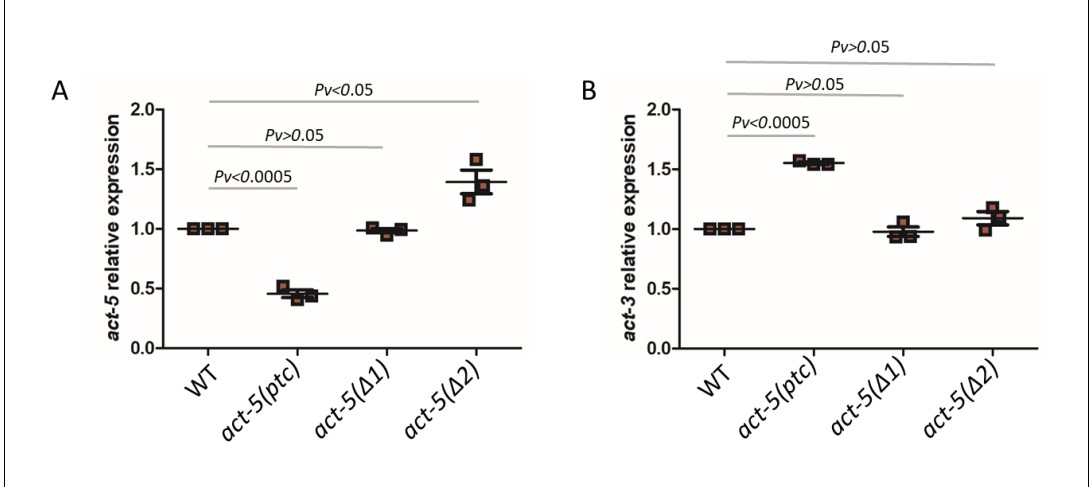

**Figure 1.** mRNA levels of *act-5* and *act-3* in WT and mutant alleles. qPCR analysis of *act-5* (**A**) and *act-3* (**B**) mRNA levels in WT and *act-5(ptc)*, *act-5(Δ1)*, and *act-5(Δ2)* mutants. *act-3* mRNA levels are upregulated when *act-5* mutant mRNA levels are reduced (i.e., only in the *act-5(ptc)* allele). WT expression levels are set at 1. Data are mean ± S.E.M.; average dCt values are shown in ***Figure 1—source data 1***. Two-tailed Student's t-test was used to calculate *P* values.

The online version of this article includes the following source data and figure supplement(s) for figure 1:

**Source data 1.** Average dCt values from qPCR analysis of *act-5* and *act-3* mRNA levels in WT and *act-5* mutants.
**Source data 2.** Average dCt values from qPCR analysis of *act-1*, *act-2* and *act-4* mRNA levels in WT and *act-5(ptc)* mutants.
**Figure supplement 1.** Organization of *act-5* locus.
**Figure supplement 2.** mRNA levels of *act-1* (**A**), *act-2* (**B**) and *act-4* (**C**) in WT and *act-5(ptc)* mutants.
**Figure supplement 3.** Pre-mRNA levels of *act-3* in WT and *act-5(ptc)* mutants.

mutants represents a model of transcriptional adaptation, we analyzed another *act-5* deletion allele (*ok1397*) (***Estes et al., 2011***), hereafter referred to as *act-5(Δ2)*. The *ok1397* deletion removes part of the promoter region and the first two exons (***Figure 1—figure supplement 1***). We examined this allele for the presence of any transcripts and identified a new isoform which is present in mutants but not in wild type (see Materials and methods) and consists of only 3' sequence (***Figure 1—figure supplement 1***). As with the *act-5(Δ1)* deletion allele, we found no changes in *act-5* or *act-3* mRNA levels in *act-5(Δ2)* mutants compared to wild type (***Figure 1B***).

In multicellular organisms, paralogous genes are often expressed in distinct spatiotemporal patterns, an indication of subfunctionalization (***Guschanski et al., 2017***). For example, in *C. elegans*, *act-3* is expressed in the pharynx (***Hunt-Newbury et al., 2007***) while *act-5* is expressed in intestinal cells (***MacQueen et al., 2005***). The models of transcriptional adaptation suggest a cell-autonomous mechanism, that is the mutant mRNA can cell-autonomously trigger transcriptional adaptation. In order to test this hypothesis, we generated transcriptional reporter constructs with the *act-3* or *act-5* promoter region driving the expression of a red florescent protein gene (***Merzlyak et al., 2007***). As expected, we observed expression of the extrachromosomal *act-5p::rfp* transgene in the intestine in wild-type animals (***Figure 2A***) as well as in *act-5(ptc)* mutants (***Figure 2—figure supplement 1***). Likewise, expression of the extrachromosomal *act-3p::rfp* transgene was only observed in the pharynx in wild-type animals (***Figure 2B***), consistent with previous studies (***Hunt-Newbury et al., 2007***). However, extrachromosomal *act-3p::rfp* expression was also observed in the intestine in *act-5(ptc)* mutants (***Figure 2C***), consistent with transcriptional adaptation. In summary, we saw upregulation of expression from a synthetic and extrachromosomal *act-3* promoter in tissues where *act-5* is expressed, supporting the model that the mutant mRNA cell-autonomously triggers transcriptional adaptation.

To identify an additional example of transcriptional adaptation in *C. elegans*, we turned to the *titin* gene family (***Figure 3—source data 1***). Due to their size, *titin* genes are frequent targets of random mutagenesis, and several PTC alleles have been identified (***Jorgensen and Mango, 2002***; ***Lipinski et al., 2011***). We focused on *unc-89* which has many nonsense alleles that do not exhibit an obvious phenotype, potentially indicating functional compensation. We identified three different

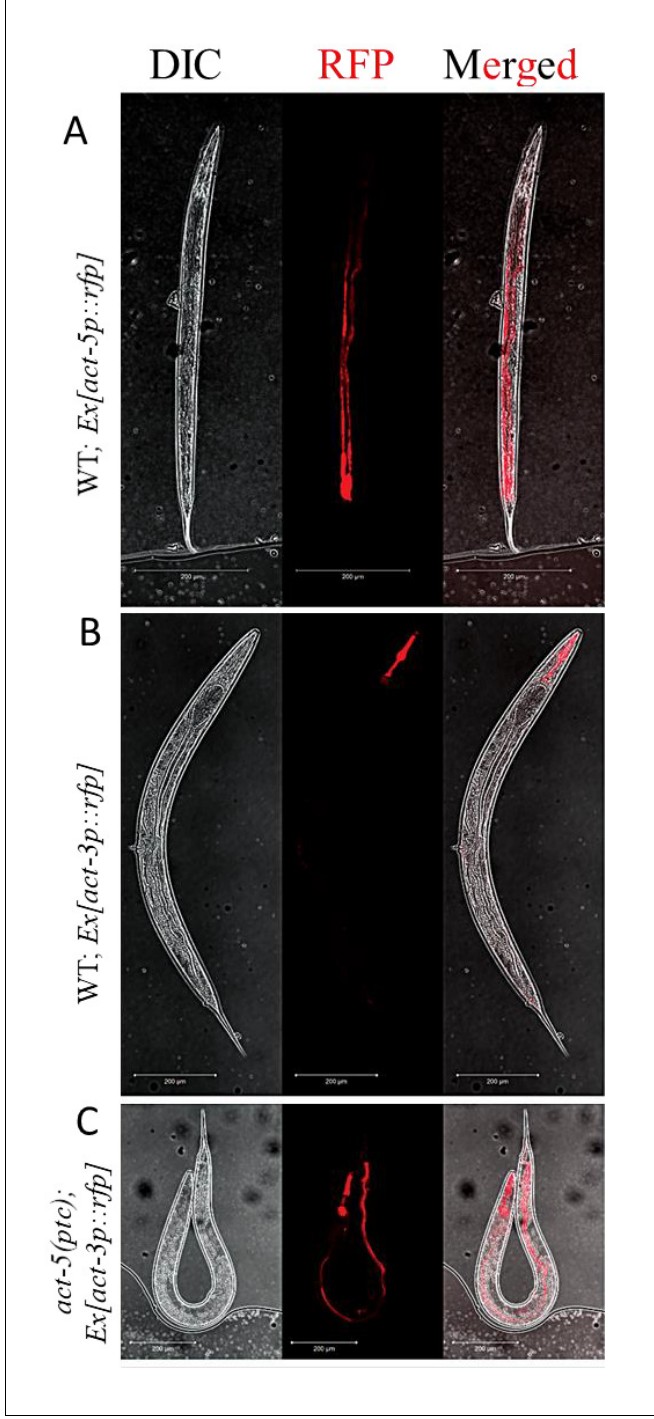

**Figure 2.** Extrachromosomal reporter expression in WT and mutant alleles. (**A**) *act-5p::rfp* extrachromosomal reporter expression was observed in the intestine in 153 of 300 WT animals. (**B**) *act-3p::rfp* extrachromosomal reporter expression was observed in the pharynx in 182 of 400 WT animals. (**C**) *act-3p::rfp* extrachromosomal reporter expression was observed in the pharynx and intestine in 138 of 320 *act-5(ptc)* mutants.

The online version of this article includes the following figure supplement(s) for figure 2:

**Figure supplement 1.** *act-5p::rfp* extrachromosomal reporter expression.

unc-89 alleles (gk469156, gk509355, gk506355) which exhibit lower levels of mutant mRNA compared to wild type (Figure 3A) and lack an obvious phenotype. Analyzing the mRNA levels of 10 titin related genes (him-4, ttn-1, ketn-1, sax-3, unc-22, unc-52, sax-7, rig-6, unc-40, and unc-73), we found that sax-3 was upregulated in all three unc-89(ptc) alleles (Figure 3—figure supplement 1), both at the mRNA (Figure 3B) and pre-mRNA (Figure 3—figure supplement 2) levels. To test whether this upregulation of sax-3 was due to transcriptional adaptation and not to the loss of UNC-89 function, we generated a 16 kb deletion (bns7000) in unc-89, hereafter referred to as unc-89(Δ), using CRISPR/Cas9 genome editing. This deletion removes part of the promoter region and the first several exons (Figure 3—figure supplement 1). Hence, most unc-89 isoforms are not observed in unc-89(Δ) mutants (Figure 3A). Homozygous unc-89(Δ) worms are maternal-effect sterile and exhibit growth defects, phenotypes not observed in the unc-89(ptc) alleles. In the RNA-less unc-89(Δ) allele, sax-3 was not upregulated (Figure 3B), indicating that sax-3 upregulation is not due to the loss of UNC-89 function and that the mutant mRNA needs to be present for the transcriptional adaptation response. Thus, sax-3 upregulation in the unc-89(ptc) alleles is a second example of transcriptional adaptation in C. elegans.

To test whether the observed changes in gene expression in act-5(ptc) and unc-89(ptc) mutants were specific, we measured unc-89 and sax-3 expression in act-5(ptc) mutants as well as act-5 and act-3 expression in unc-89(ptc) mutants. We observed no significant differences (Figure 3—figure supplement 3), suggesting that there is specificity to the gene expression changes.

## Identifying additional regulators of transcriptional adaptation

The mutant mRNA has been reported to activate transcriptional adaptation in zebrafish embryos and mouse cell lines (El-Brolosy et al., 2019; Ma et al., 2019). In order to identify additional factors involved in transcriptional adaptation, we performed a candidate RNA interference (RNAi) screen,

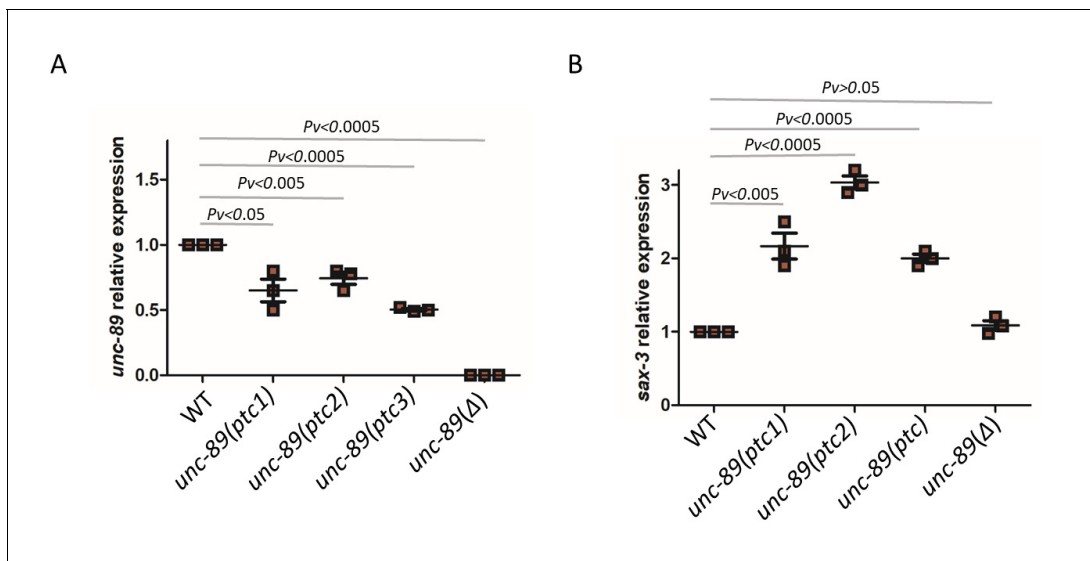

**Figure 3.** mRNA levels of unc-89 and sax-3 in WT and mutant alleles. qPCR analysis of unc-89 (C) and sax-3 (D) mRNA levels in WT and unc-89(ptc1), unc-89(ptc2), unc-89(ptc3), and unc-89(Δ) mutants. sax-3 mRNA levels in unc-89 alleles are upregulated when unc-89 mutant mRNA levels are reduced, except in the deletion allele. WT expression levels are set at 1. Data are mean ± S.E.M.; average dCt values are shown in Figure 3—source data 2. Two-tailed Student's t-test was used to calculate P values.

The online version of this article includes the following source data and figure supplement(s) for figure 3:

**Source data 1.** List of ttn-1 paralogous genes based on WormBase release WS266.
**Source data 2.** Average dCt values from qPCR analysis of unc-89 and sax-3 mRNA levels in WT and unc-89 mutants.
**Source data 3.** Distance, in nucleotides, from each PTC to the next exon-intron junction and to the stop codon in each unc-89 isoform in the unc-89(ptc1), unc-89(ptc2), and unc-89(ptc3) alleles.
**Figure supplement 1.** Organization of unc-89 locus.
**Figure supplement 2.** Pre-mRNA levels of sax-3 in WT and unc-89(ptc1), unc-89(ptc2), unc-89(ptc3) mutants.
**Figure supplement 3.** mRNA levels in WT and mutant alleles.

focusing on genes involved in RNA metabolism (*Figure 4—source data 1*). We knocked down genes involved in mRNA processing including splicing and nonsense-mediated decay, as well as other genes involved in small RNA synthesis and maturation. We measured the mRNA levels of the mutant and adapting genes in order to position RNAi candidates upstream or downstream of mRNA decay. If the gene targeted by RNAi is required for mutant mRNA decay, we expect to see the mRNA levels of the mutant and adapting genes to be similar to wild-type levels. If the gene targeted by RNAi is involved in transcriptional adaptation downstream of mutant mRNA decay, we expect to see the levels of mutant mRNA remaining lower than in wild type, but the levels of the adapting gene's mRNA to be similar to wild-type levels. Finally, if the gene targeted by RNAi is not involved in transcriptional adaptation, we expect to see the levels of mutant mRNA remain lower than in wild type and the expression levels of the adapting gene to remain higher than in wild type. For example, when we knocked down *drsh-1*, a gene involved in miRNA biogenesis (*Denli et al., 2004*), we saw no significant changes in the mRNA levels of the mutant or adapting genes compared to control (*Figure 4—figure supplement 1A,B,C,D*), suggesting that *drsh-1* is not involved in regulating transcriptional adaptation.

Transcriptional adaptation requires the activity of decay factors (*El-Brolosy et al., 2019*; *Ma et al., 2019*), and UPF1, SMG6, and XRN1 were reported to be differentially required in various zebrafish embryo and mouse cell line models of transcriptional adaptation (*El-Brolosy et al., 2019*). In order to test whether NMD factors are involved in regulating transcriptional adaptation in *C. elegans*, we knocked down several NMD genes including *smg-2* (the *C. elegans* orthologue of *Upf1*), *smg-4* (*Upf3*) and *smg-6* (*Smg6*). Knockdown of *smg-2* and *smg-4* blocked the transcriptional adaptation response in all three *unc-89(ptc)* alleles but not in the *act-5(ptc)* allele (*Figure 4*). Conversely, knockdown of *smg-6* blocked the transcriptional adaptation response in the *act-5(ptc)* allele but not in the three *unc-89(ptc)* alleles (*Figure 4*). A differential requirement for Upf1 and Smg6 between gene models was also observed in mouse cells (*El-Brolosy et al., 2019*).

As RNAi efficiency can vary in different tissues (*Ratliff et al., 2006*; *Zhuang and Hunter, 2011*), we generated double mutant strains with *smg-2*, *smg-4*, or *smg-6* mutant alleles and the *act-5(ptc)* and *unc-89(ptc)* alleles to exclude the possibility of tissue-specific knockdown. Analysis of the double mutant strains confirmed the observations made in the RNAi experiments (*Figure 5*). For example, we found that the levels of *act-5* mRNA were lower in *smg-2; act-5(ptc)* and *smg-4; act-5(ptc)* double mutants than in *smg-2* and *smg-4* single mutants, and that the levels of the adapting gene's mRNA were higher (*Figure 5—source data 1*), further indicating that *smg-2* and *smg-4* are not required for transcriptional adaptation in the *act-5* model. However, in *smg-4; unc-89(ptc)* animals, the mRNA levels of the mutant (*unc-89*) and adapting (*sax-3*) genes were similar to those in *smg-4* single mutants. Furthermore, these animals exhibited a mild uncoordinated phenotype and grew slowly, suggesting a lack of functional compensation. These data further indicate that *smg-4* is required for transcriptional adaptation in the *unc-89* model. We could not obtain *smg-6; act-5(ptc)* viable mutants due to severe larval lethality, possibly as a consequence of blocking the transcriptional adaptation response, that is *act-3* upregulation. *smg-6; unc-89(ptc)* mutants exhibited lower levels of *unc-89* mRNA and higher levels of adapting gene mRNA in comparison to single *smg-6* mutants (*Figure 5—source data 1*) similar to the observations in the RNAi experiments. Thus, there are differential requirements for decay factors in different models of transcriptional adaptation.

Previous data indicate that the exonuclease Xrn1 is involved in regulating the transcriptional adaptation response in mouse cells (*El-Brolosy et al., 2019*). Therefore, we tested the role of exonucleases in transcriptional adaptation in *C. elegans*, specifically the exonuclease gene *xrn-1* (*Jones et al., 2012*) and the XRN-2 partner gene *paxt-1* (*Miki et al., 2014*) (*Figure 4*). We found that knocking down *xrn-1* or *paxt-1* led to mutant (*act-5* and *unc-89*) mRNA levels similar to wild-type levels. Furthermore, the transcriptional adaptation response was blocked, suggesting that the degradation and processing of mutant transcripts is important to trigger transcriptional adaptation.

We next looked for additional factors required for transcriptional adaptation. Pre-mRNA splicing and NMD are closely related processes via the positioning and use of the exon junction complex (EJC) (*Lejeune and Maquat, 2005*; *Kashima et al., 2010*; *Fukumura et al., 2016*). SR-protein kinases (SRPK) and their substrates, serine/arginine-rich (SR) splicing factors, are key components of the splicing machinery and are well conserved across phyla (*Kuroyanagi et al., 2000*; *Black, 2003*; *Galvin et al., 2011*). Multiple SR proteins are components of the EJC (*Singh et al., 2012*), consistent with a previously suggested role of SR proteins in mRNA surveillance (*Zhang and Krainer, 2004*).

| process | gene | act-5/act-3 | | unc-89/sax-3 | |
|---|---|---|---|---|---|
| | | mutant mRNA reduction | upregulation of adapting gene | mutant mRNA reduction | upregulation of adapting gene |
| | control | Yes | Yes | Yes | Yes |
| mRNA splicing | spk-1 | No | No | No | No |
| | rsp-6 | No | No | Yes | Yes |
| mRNA decay and processing | smg-2 | Yes | Yes | No | No |
| | smg-4 | Yes | Yes | No | No |
| | smg-6 | No | No | Yes | Yes |
| | xrn-1 | No | No | No | No |
| | paxt-1 | No | No | No | No |
| small RNA biogenesis | ergo-1 | Yes | No | Yes | No |
| | nrde-3 | Yes | No | Yes | No |
| | rrf-3 | Yes | No | Yes | No |
| | dcr-1 | Yes | No | Yes | No |

**Figure 4.** Factors regulating transcriptional adaptation identified in RNAi-mediated knockdown screen.
The online version of this article includes the following source data and figure supplement(s) for figure 4:

**Source data 1.** List of genes and RNAi clones tested in the screen; average dCt values of qPCR analyses of act-5 and act-3 mRNA levels in WT and act-5 mutants as well as of unc-89 and sax-3 mRNA levels in WT and unc-89 mutants.

**Figure supplement 1.** qPCR analysis of act-5 (A) and act-3 (B) mRNA levels in WT and act-5(ptc) mutants as well as of unc-89 (C) and sax-3 (D) mRNA levels in WT and unc-89(ptc) mutants upon drsh-1 RNAi-mediated knockdown by two independent clones.

**Figure supplement 2.** qPCR analysis of act-5 (A) and act-3 (B) mRNA levels in WT and act-5(ptc) mutants as well as of unc-89 (C) and sax-3 (D) mRNA levels in WT and unc-89(ptc) mutants upon spk-1 RNAi-mediated knockdown by two independent clones.

*Figure 4 continued on next page*

*Figure 4 continued*

**Figure supplement 3.** qPCR analysis of *act-5* (**A**) and *act-3* (**B**) mRNA levels in WT and *act-5(ptc)* mutants as well as of *unc-89* (**C**) and *sax-3* (**D**) mRNA levels in WT and *unc-89(ptc)* mutants upon *nrde-3* RNAi-mediated knockdown by two independent clones.

Knocking down the SRPK gene *spk-1* resulted in mutant mRNA levels similar to wild-type levels, and blocked transcriptional adaptation in all *act-5* and *unc-89 ptc* alleles (***Figure 4—figure supplement 2A,B,C,D***). We also identified the SR family gene *rsp-6* as a regulator of transcriptional adaptation in the *act-5* model, but were unable to identify a single SR protein whose knockdown influenced the transcriptional adaptation response in the *unc-89* model (***Figure 4—source data 1***), possibly due in part to the complexity of the *unc-89* gene structure including the large number of isoforms (***Tourasse et al., 2017***).

The next group of genes we targeted encode factors involved in small RNA (sRNA) biogenesis, maturation and transport into the nucleus. We tested several pathways (***Figure 4—source data 1***) and observed that the argonaute proteins ERGO-1 and NRDE-3, the RNA-dependent RNA polymerase RRF-3, as well as the ribonuclease DCR-1 regulate the transcriptional adaptation response downstream of mRNA decay (*i.e.*, the mutant mRNA was still degraded but the adapting gene was not upregulated) (***Figure 4***, ***Figure 4—figure supplement 3A,B,C,D***, ***Figure 4—source data 1***). These RNAi data were confirmed by analyzing double mutants of the *act-5(ptc)* or *unc-89(ptc)* alleles with *ergo-1*, *nrde-3*, and *rrf-3* (***Figure 5***), and all these animals exhibited phenotypes comparable to the

| process | gene (allele) | *act-5/act-3* | | *unc-89/sax-3* | |
|---|---|---|---|---|---|
| | | mutant mRNA reduction | upregulation of adapting gene | mutant mRNA reduction | upregulation of adapting gene |
| mRNA decay | *smg-2(e2008)* | Yes | Yes | not tested | not tested |
| | *smg-4(ma116)* | Yes | Yes | No | No |
| | *smg-6(r896)* | not tested | not tested | Yes | Yes |
| small RNA biogenesis | *ergo-1(gg100)* | Yes | No | Yes | No |
| | *nrde-3(gg66)* | Yes | Yes | Yes | No |
| | *rrf-3(mgm373)* | Yes | No | Yes | No |

**Figure 5.** Factors regulating transcriptional adaptation analyzed in double mutants.
The online version of this article includes the following source data and figure supplement(s) for figure 5:

**Source data 1.** List of genes and alleles for each gene tested in the double mutant analysis; average dCt values from qPCR analyses of *act-5* and *act-3* mRNA levels in WT and *act-5* mutants as well as of *unc-89* and *sax-3* mRNA levels in WT and *unc-89* mutants.
**Figure supplement 1.** Partial data from double mutant analysis.

*act-5* or *unc-89* deletion alleles analyzed in this study including larval lethality, slow growth and unco-ordinated movements, indicating lack of functional compensation. Notably, ERGO-1, NRDE-3, RRF-3, and DCR-1 are involved in 26G RNA biogenesis (*Pavelec et al., 2009*; *Vasale et al., 2010*; *Fischer, 2010*; *Grishok, 2013*; *Yvert, 2014*), suggesting that 26G RNAs could play a role in transcriptional adaptation.

Together, these results indicate that mRNA decay as well as small RNA biogenesis and transport are critical in triggering transcriptional adaptation.

## Discussion

Recent advances in reverse genetic tools have significantly expanded our ability to generate genetic modifications in a wide range of organisms (*Housden et al., 2017*). However, some engineered mutants exhibit no apparent phenotype, renewing interest in the concept of genetic robustness. Genetic compensation, and in particular transcriptional adaptation, have been proposed as a means to achieve genetic robustness upstream of protein feedback loops. Despite the potential importance of transcriptional adaptation, its underlying molecular mechanisms remain relatively unexplored. Here, we report two cases of transcriptional adaptation in *C. elegans*. By carrying out a small RNAi screen and a follow up analysis using double mutants, we identified several new factors that regulate transcriptional adaptation and further validated previously identified ones.

In the *C. elegans act-5* model, the mutant gene and related adapting gene (*act-5* and *act-3*, respectively) are primarily expressed in distinct tissues. However, using an extrachromosomal transcriptional reporter, we observed that in *act-5(ptc)* mutants, the *act-3* promoter adapts to drive transcription in the primary site of *act-5* expression, the intestine. As *act-5(ptc)* mutants do not exhibit any obvious phenotype when the transcriptional adaptation response is intact, we predict that ACT-3 and/or other proteins are able to compensate for the loss of ACT-5. Indeed, when we disrupted transcriptional adaptation, *act-5(ptc)* mutants did not survive. Thus, transcriptional adaptation can in some cases entail the change in the pattern of expression of related gene(s) and suppress phenotypes that would alter the animal's fitness.

Based on the factors identified in this study, we hypothesize that the transcriptional adaptation response consists of at least three critical processes: mutant mRNA decay, sRNA maturation and sRNA transport. In terms of mutant mRNA decay, we found that the machinery can be gene-specific. In our experiments, SMG-6 is involved in *act-5(ptc)* mRNA decay, while SMG-2 (UPF1) and SMG-4 (UPF3) impact *unc-89(ptc)* mRNA decay. Similar observations were made in mouse *Actb* and *Rela* mutant cells in which siRNA-mediated knock down of SMG6 blocked the transcriptional adaptation response in *Actb* mutant mESCs but had little influence on *Rela* mutant MEFs. Conversely, siRNA-mediated knockdown of UPF1 blocked the transcriptional adaptation response in *Rela* mutant MEFs but not in *Actb* mutant mESCs (*El-Brolosy et al., 2019*). Consistently, mutant mRNA decay can involve different factors in the same organism (*Nickless et al., 2017*), possibly due to differential expression of the decay factors. However, we cannot exclude the possibility that SMG6 could function as a decay factor independent of NMD, especially since it has been reported to have NMD-independent cleavage activity (*Gehring et al., 2005*; *Glavan et al., 2006*; *Huntzinger et al., 2008*; *Chakrabarti et al., 2014*).

Transcriptional adaptation can be triggered by the degradation products of the mutant mRNA (*El-Brolosy et al., 2019*), which could seed the generation of sRNAs (*Mattick and Makunin, 2005*; *Boivin et al., 2018*). We found that factors involved in sRNA maturation and transport, including RRF-3, DCR-1, ERGO-1 and NRDE-3, also regulate transcriptional adaptation. Transcriptional modulation of genes by sRNAs of approximately 20–30 nucleotides in length is a widespread and diverse feature of prokaryotes (*Melamed et al., 2019*) and eukaryotes (*Ambros et al., 2003*; *Yigit et al., 2006*; *Hutvagner and Simard, 2008*; *Portnoy et al., 2011*; *Castel and Martienssen, 2013*; *Rechavi and Lev, 2017*; *Billmyre et al., 2019*). Notably, the factors we identified are known to be involved in somatic gene regulation by sRNAs, described as the RRF-3 pathway (*Gent et al., 2010*). RRF-3 is an RNA-dependent RNA polymerase involved along with the DICER complex in the biogenesis of 26-nucleotide RNAs with 5' bias for guanosine monophosphate (26G-RNAs) (*Han et al., 2009*; *Gent et al., 2010*; *Vasale et al., 2010*). 26G sRNAs associate with the Argonaute protein ERGO-1, which is involved in the further maturation of sRNAs and is required to separate the sRNA duplex (*Han et al., 2009*; *Gent et al., 2010*; *Fischer et al., 2011*). Mature sRNAs interacting with

Argonaute proteins can direct post-transcriptional gene silencing (*Vasale et al., 2010*; *Phillips et al., 2012*), or be transported into the nucleus (*Guang et al., 2008*; *Buckley et al., 2012*; *Shirayama et al., 2012*). NRDE-3 is another Argonaute protein involved in transporting sRNAs into the nucleus (*Guang et al., 2008*), and we found that knocking it down, and knocking it out, blocked transcriptional adaptation while not affecting mutant mRNA levels. While sRNAs are best known as repressors of gene expression, they can also function as activators (*Li et al., 2006*; *Janowski et al., 2006*; *Turunen et al., 2009*; *Portnoy et al., 2011*; *Wedeles et al., 2013*; *Li, 2017*), although the underlying mechanisms remain poorly understood (*Portnoy et al., 2011*). Some of these activating sRNAs interact with Argonaute proteins (*Seth et al., 2013*), and they can target gene regulatory sequences including promoters. Whether they can also interfere with antisense RNAs, which usually function to repress gene expression (*Faghihi and Wahlestedt, 2009*; *Modarresi et al., 2012*), is a hypothesis worth testing given our observations in zebrafish embryos and mouse cell lines (*El-Brolosy et al., 2019*) as well as the previously suggested role of Argonaute proteins in such a process (*Ghanbarian et al., 2017*).

The transcriptional adaptation factors identified here came from a candidate screen where we specifically targeted pathways involved in RNA metabolism. With this study, we have established *C. elegans* as a genetic model system to perform unbiased screens to help reveal further mechanisms of transcriptional adaptation, a newly uncovered phenomenon contributing to genetic robustness.

# Materials and methods

**Key resources table**

| Reagent type (species) or resource | Designation | Source or reference | Identifiers | Additional information |
|---|---|---|---|---|
| Gene (*Caenorhabditis elegans*) | *act-1* | | CELE_T04C12.6 | WormBase ID: WBGene00000063 |
| Gene (*Caenorhabditis elegans*) | *act-2* | | CELE_T04C12.5 | WBGene00000064 |
| Gene (*Caenorhabditis elegans*) | *act-3* | | CELE_T04C12.4 | WBGene00000065 |
| Gene (*Caenorhabditis elegans*) | *act-4* | | CELE_M03F4.2 | WBGene00000066 |
| Gene (*Caenorhabditis elegans*) | *act-5* | | CELE_T25C8.2 | WBGene00000067 |
| Gene (*Caenorhabditis elegans*) | *unc-89* | | CELE_C09D1.1 | WBGene00006820 |
| Gene (*Caenorhabditis elegans*) | *sax-3* | | CELE_ZK377.2 | WBGene00004729 |
| Strain, strain background (*C. elegans*) | N2 | CGC, Bristol strain | | wild type |
| Strain, strain background (*C. elegans*) | IN2049 | *MacQueen et al., 2005* | | *act-5(ptc); dtIs 419[act-5+ rol-6(d)]* |
| Strain, strain background (*C. elegans*) | IN2051 | *MacQueen et al., 2005* | | *act-5(Δ1); dtIs 419[act-5+ rol-6(d)]* |
| Strain, strain background (*C. elegans*) | VC971 | CGC, *Estes et al., 2011* | | *+/mT1; act-5(Δ2)/ mT1 [dpy-10(e128)].* |
| Strain, strain background (*C. elegans*) | CB4043 | CGC, *Hodgkin et al., 1989* | | *smg-2(e2008);him-5(e1490)* |
| Strain, strain background (*C. elegans*) | CB4355 | CGC, *Pulak and Anderson, 1993* | | *smg-4(ma116);him-8(e1490)* |
| Strain, strain background (*C. elegans*) | TR1396 | CGC, *Pulak and Anderson, 1993* | | *smg-6(r896)* |
| Strain, strain background (*C. elegans*) | YY168 | CGC, *Pavelec et al., 2009* | | *ergo-1(gg100)* |

*Continued on next page*

*Continued*

| Reagent type (species) or resource | Designation | Source or reference | Identifiers | Additional information |
|---|---|---|---|---|
| Strain, strain background (*C. elegans*) | YY158 | CGC, *Guang et al., 2008* | | *nrde-3(gg66)* |
| Strain, strain background (*C. elegans*) | YY13 | CGC, *Pavelec et al., 2009* | | *rrf-3(mg373)* |
| Strain, strain background (*C. elegans*) | DYS0005 | This study, crossed IN2049 to N2 | | *act-5(ptc)* |
| Strain, strain background (*C. elegans*) | DYS0004 | This study, crossed IN2049 to N2 | | *+/act-5(Δ1)* |
| Strain, strain background (*C. elegans*) | DYS0012 | This study, injected in N2 | | *Ex[act-5p::RFP]* |
| Strain, strain background (*C. elegans*) | DYS0014 | This study, injected in N2 | | *Ex[act-3p::RFP]* |
| Strain, strain background (*C. elegans*) | DYS0015 | This study, crossed DYS0014 to DYS0004 | | *act-5(ptc);Ex[act-3p::RFP]* |
| Strain, strain background (*C. elegans*) | DYS0042 | This study, crossed DYS0012 to DYS0005 | | *act-5(ptc);Ex[act-5p::RFP]* |
| Strain, strain background (*C. elegans*) | VC40114 | CGC, Million Mutation Project | | *unc-89(ptc1)* |
| Strain, strain background (*C. elegans*) | VC40193 | CGC, Million Mutation Project | | *unc-89(ptc2)* |
| Strain, strain background (*C. elegans*) | VC40199 | CGC, Million Mutation Project | | *unc-89(ptc3)* |
| Strain, strain background (*C. elegans*) | DYS0028 | This study, crossed VC40114 to N2 | | *unc-89(ptc1)* |
| Strain, strain background (*C. elegans*) | DYS0030 | This study, crossed VC40193 to N2 | | *unc-89(ptc2)* |
| Strain, strain background (*C. elegans*) | DYS0031 | This study, crossed VC40199 to N2 | | *unc-89(ptc3)* |
| Strain, strain background (*C. elegans*) | DYS0037 | This study, induced by CRISPR/Cas9 | | *unc-89(Δ)* |
| Strain, strain background (*C. elegans*) | DYS0008 | This study, crossed DYS0005 to CB4043 | | *smg-2(e2008); act-5(ptc)* |
| Strain, strain background (*C. elegans*) | DYS0057 | This study, crossed DYS0005 to CB4355 | | *act-5(ptc); smg-4(ma116)* |
| Strain, strain background (*C. elegans*) | DYS0047 | This study, crossed DYS0028 to CB4355 | | *unc-89(ptc1); smg-4(ma116)* |
| Strain, strain background (*C. elegans*) | DYS0048 | This study, crossed DYS0030 to CB4355 | | *unc-89(ptc2); smg-4(ma116)* |
| Strain, strain background (*C. elegans*) | DYS0050 | This study, crossed DYS0031 to CB4355 | | *unc-89(ptc3); smg-4(ma116)* |
| Strain, strain background (*C. elegans*) | DYS0053 | This study, crossed DYS0028 to TR1396 | | *unc-89(ptc1); smg-6(r896)* |
| Strain, strain background (*C. elegans*) | DYS0055 | This study, crossed DYS0030 to TR1396 | | *unc-89(ptc2); smg-6(r896)* |
| Strain, strain background (*C. elegans*) | DYS0056 | This study, crossed DYS0031 to TR1396 | | *unc-89(ptc3); smg-6(r896)* |
| Strain, strain background (*C. elegans*) | DYS0010 | This study, crossed DYS0005 to YY168 | | *act-5(ptc); ergo-1(gg100)* |
| Strain, strain background (*C. elegans*) | DYS0054 | This study, crossed DYS0028 to YY168 | | *unc-89(ptc1); ergo-1(gg100)* |
| Strain, strain background (*C. elegans*) | DYS0051 | This study, crossed DYS0030 to YY168 | | *unc-89(ptc2); ergo-1(gg100)* |

*Continued on next page*

*Continued*

| Reagent type (species) or resource | Designation | Source or reference | Identifiers | Additional information |
|---|---|---|---|---|
| Strain, strain background (*C. elegans*) | DYS0052 | This study, crossed DYS0031 to YY168 | | *unc-89(ptc3); ergo-1(gg100)* |
| Strain, strain background (*C. elegans*) | DYS0045 | This study, crossed DYS0005 to YY158 | | *act-5(ptc); nrde-3(gg66)* |
| Strain, strain background (*C. elegans*) | DYS0065 | This study, crossed DYS0028 to YY158 | | *unc-89(ptc1); nrde-3(gg66)* |
| Strain, strain background (*C. elegans*) | DYS0072 | This study, crossed DYS0030 to YY158 | | *unc-89(ptc2); nrde-3(gg66)* |
| Strain, strain background (*C. elegans*) | DYS0066 | This study, crossed DYS0031 to YY158 | | *unc-89(ptc3); nrde-3(gg66)* |
| Strain, strain background (*C. elegans*) | DYS0046 | This study, crossed DYS0005 to YY13 | | *rrf-3(mg373); act-5(ptc)* |
| Strain, strain background (*C. elegans*) | DYS0070 | This study, crossed DYS0028 to YY13 | | *unc-89(ptc1); rrf-3(mg373)* |
| Strain, strain background (*C. elegans*) | DYS0062 | This study, crossed DYS0030 to YY13 | | *unc-89(ptc2); rrf-3(mg373)* |
| Strain, strain background (*C. elegans*) | DYS0063 | This study, crossed DYS0031 to YY13 | | *unc-89(ptc3); rrf-3(mg373)* |
| Commercial assay or kit | In-Fusion HD Cloning | Clontech | Clontech:639647 | |
| Commercial assay or kit | Superscript III reverse transcriptase | Takara | Cat. No: 18080–044 | |
| Commercial assay or kit | SMARTer RACE cDNA Amplification Kit | Takara | Cat. N. 634860 | |
| Commercial assay or kit | Advantage 2 PCR kit | Takara | Cat. N. 639207 | |
| RNAi construct | mv_C18D11.4 | BioScience | | *rsp-8* |
| RNAi construct | sjj2_C18D11.4 | BioScience | | *rsp-8* |
| RNAi constructs | mv_C33H5.12 | BioScience | | *rsp-6* |
| RNAi constructs | sjj2_C33H5.12 | BioScience | | *rsp-6* |
| RNAi constructs | mv_W02B12.3 | BioScience | | *rsp-1* |
| RNAi constructs | sjj2_W02B12.3 | BioScience | | *rsp-1* |
| RNAi constructs | mv_D2089.1 | BioScience | | *rsp-7* |
| RNAi constructs | sjj2_D2089.1 | BioScience | | *rsp-7* |
| RNAi constructs | mv_B0464.5 | BioScience | | *spk-1* |
| RNAi constructs | sjj2_B0464.5 | BioScience | | *spk-1* |
| RNAi constructs | mv_R05D11.6 | BioScience | | *paxt-1* |
| RNAi constructs | sjj2_R05D11.6 | BioScience | | *paxt-1* |
| RNAi constructs | mv_F43E2.8 | BioScience | | *hsp-4* |
| RNAi constructs | sjj2_F43E2.8 | BioScience | | *hsp-4* |
| RNAi constructs | sjj2_Y39G8C.1 | BioScience | | *xrn-1* |
| RNAi constructs | mv_Y48G8AL.6 | BioScience | | *smg-2* |
| RNAi constructs | sjj2_Y48G8AL.6 | BioScience | | *smg-2* |
| RNAi constructs | sjj2_F46B6.3 | BioScience | | *smg-4* |
| RNAi constructs | mv_Y54F10AL.2 | BioScience | | *smg-6* |
| RNAi constructs | sjj2_Y54F10AL.2 | BioScience | | *smg-6* |
| RNAi constructs | mv_F26B1.2 | BioScience | | *hrpk-1* |
| RNAi constructs | sjj2_F26B1.2 | BioScience | | *hrpk-1* |

*Continued*

| Reagent type (species) or resource | Designation | Source or reference | Identifiers | Additional information |
|---|---|---|---|---|
| RNAi constructs | mv_F26E4.10 | BioScience | | *drsh-1* |
| RNAi constructs | sjj2_F26E4.10 | BioScience | | *drsh-1* |
| RNAi constructs | mv_T22A3.5 | BioScience | | *pash-1* |
| RNAi constructs | sjj2_T22A3.5 | BioScience | | *pash-1* |
| RNAi constructs | sjj2_F26A3.8 | BioScience | | *rrf-1* |
| RNAi constructs | mv_ R06C7.1 | BioScience | | *wago-1* |
| RNAi constructs | sjj2_ R06C7.1 | BioScience | | *wago-1* |
| RNAi constructs | mv_F58G1.1 | BioScience | | *wago-4* |
| RNAi constructs | sjj2_F58G1.1 | BioScience | | *wago-4* |
| RNAi constructs | sjj2_F10B5.7 | BioScience | | *rrf-3* |
| RNAi constructs | mv_M88.5 | BioScience | | *zbp-1* |
| RNAi constructs | sjj2_M88.5 | BioScience | | *zbp-1* |
| RNAi constructs | sjj2_K12H4.8 | BioScience | | *dcr-1* |
| RNAi constructs | mv_T20G5.11 | BioScience | | *rde-4* |
| RNAi constructs | sjj2_T20G5.11 | BioScience | | *rde-4* |
| RNAi constructs | mv_F36H1.2 | BioScience | | *kdin-1* |
| RNAi constructs | mv_K12B6.1 | BioScience | | *sago-1* |
| RNAi constructs | sjj2_K12B6.1 | BioScience | | *sago-1* |
| RNAi constructs | mv_K08H10.7 | BioScience | | *rde-1* |
| RNAi constructs | sjj2_K08H10.7 | BioScience | | *rde-1* |
| RNAi constructs | sjj2_R09A1.1 | BioScience | | *ergo-1* |
| RNAi constructs | mv_R04A9.2 | BioScience | | *nrde-3* |
| RNAi constructs | sjj2_R04A9.2 | BioScience | | *nrde-3* |

## Culture conditions and strains

All wild-type worms were the N2 reference strain. All *C. elegans* strains were kept on 6 cm plates with nematode growth medium agar and fed with a lawn of *E. coli* OP50 grown in 500 µl Luria broth, except for the RNAi mediated knockdown experiments where the worms were fed with *E. coli* expressing the respective double-stranded RNA. Cultures were maintained at 20˚C. Also, to minimize the potential for laboratory evolution of the trait, a new culture of the strains was revived annually from frozen stocks. All plates with fungal or bacterial contamination were excluded from the experiments.

## Synchronization of cultures for RNA isolation

Worms from healthy cultures were washed off of plates using M9 buffer and passed through a 41 µm filter (Millipore Cat. No SCNY00040) with vacuum; antibiotics (Ampicillin, Chloramphenicol) were added (50 µg/ml final concentration) to eliminate remaining food bacteria, and the worms were then incubated on a shaker at room temperature for 15 min. Worms were centrifuged at 3000 rpm for 5 min to pellet early larval stage animals. The buffer was aspirated and 1 ml of fresh buffer was added to resuspend the pellet. Samples were confirmed to be primarily L1 and L2 stage larvae by observing two 5 µl samples on a 6 cm nematode growth medium plate. Starving cultures or cultures that had more than one male were excluded from the experiments.

## qPCR analysis

Total RNA from synchronized cultures or manually picked young adults was isolated using TRIzol (ambion by Takara). For reverse transcription (RT), Superscript III reverse transcriptase (Invitrogen, Cat. No: 18080–044) was used following manufacturer's instructions. We used 1–2 µg total RNA for

each RT reaction. The qPCR experiments were performed on a CFX Connect Real-Time System (Bio-rad-Roche Diagnostics) as described previously (*El-Brolosy et al., 2019*). *cdc-42* and *Y45F10D.4* (*iscu-1*) were used as reference genes as described previously (*Hoogewijs et al., 2008*), and the Ct values ranged from 12.3 to 28.4 for *cdc-42* and 11.8 to 26 for *Y45F10D.4*. The Ct values for all other genes were aimed to be below 30.

The following primers were used to amplify the cDNA of target genes: *Y45F10D.4* (forward 5'-CGAGAACCCGCGAAATGTCGGA-3' and reverse 5'- CGGTTGCCAGGGAAGATGAGGC-3'), *cdc-42* (forward 5'-AGCCATTCTGGCCGCTCTCG-3' and reverse 5'- GCAACCGCTTCTCGTTTGGC-3'), *act-1* (forward 5'-ACGACGAGTCCGGCCCATCC-3' and reverse 5'-GAAAGCTGGTGGTGACGATGGTT-3'), *act-2* (forward 5'-GCGCAAGTACTCCGTCTGGATCG-3' and reverse 5'- GGGTGTGAAAATCCG TAAGGCAGA-3'), *act-3* (forward 5'-AAGCTCTTCGCCTTACCATTTTCTC-3' and reverse 5'-ACA-GAGCAAATTGTAGTGGGGTCTTC-3'), *act-4* (forward 5'-AGAGGCTCTCTTCCAGCCATCCTTC-3' and reverse 5'-TGATCTTGATCTTCATGGTGGATGG-3'), *act-5* (forward 5'- AAGTGCGATGTCGACA TCAGAAAG-3' and reverse 5'- TAATCTTGATCTTCATTGTGCTTGG-3'), *act-5d* (forward 5'- AAG TGCGATGTCGACATCAGAAAG-3' and reverse 5'- TAATCTTGATCTTCATTGTGCTCCGG-3'), *unc-89* (forward 5'-AAGGCTGAACTTGTCATCGAAGGAG-3' and reverse 5'-TCATCTCCACAACA TTACCCTCGTG-3'), *sax-3* (forward 5'-TGCCGTTTGTCCCGTAACAACTATG-3' and reverse 5'-ATC TTCTGAAGCTGACGGGGAGAAC-3'), *act-3* pre-mRNA (forward 5'-TTTTTCAGAACCATGAAGA TCA-3' and reverse 5'-GAAAATGGTAAGGCGAAGAGC-3'), *sax-3* pre-mRNA (forward 5'-TG TAAAACCGCACTGCACAAT-3' and reverse 5'-TCCACCAAGAGCCTGAAAAC-3'). PCR efficiency was determined using external standards on plasmid mini-preparation of cloned PCR products. Expression levels were analyzed by basic relative quantification. qPCR data are based on three biological replicates and three technical replicates for each biological replicate.

## Rapid amplification of cDNA ends (RACE)
Total RNA from manually picked young wild-type and *act-5(Δ2)* mutant adults was isolated using TRIzol (ambion by Takara). 5' and 3' RACE ready cDNA was synthesized by reverse transcription PCR using a SMARTer RACE cDNA Amplification Kit following manufacturer's protocol (Cat. N. 634860, Takara). PCRs were performed using an Advantage 2 PCR kit (Cat. N. 639207, Takara). The following gene-specific primers and nested gene-specific primers were used to amplify 3' and 5' cDNA ends: act5GSP2 (5'-ACCACCGGAATCGTTTTGGACACCGGAG-3'), act5NGSP2 (5'-GAAGGATATGCCC TCCCACATGCCATCC-3'), act5GSP1 (5'-AAAAATCAGCTTAGAAGCACTTTCGGTG-3'), act5NGSP1 (5'-TCGATGGGCCGGACTCGTCGTACTCCTG-3'), unc89GSP2 (5'-TTTGGTACCATTTGTA TAGAGGCGAGTG-3'), unc89NGSP2 (5'-TTCTGAACTGGACAAATCTTGCTTTTCG-3), unc89N1GSP2 (5'- ACTTTCCAGTATCTCCTGGATGTTGCTTC-3'), and unc89N2GSP2 (5'- TTTGAATACTTTTTGA TGAACCGTGTGC-3'). RACE experiment revealed an isoform with an alternative start which is present only in *act-5(Δ2)* mutants (*Figure 1—figure supplement 1A*). This new isoform is not affected by the large deletion, and thus the corresponding mRNA is not degraded (*Figure 1A*).

## Plasmid construction and genetic transformation
To study the expression of *act-5*, we generated a reporter construct with an *act-5* promoter region (2.5 kb from III:13606066 to 13608569) fused to turboRFP in a pUC19 vector. Similarly, a pUC19 vector containing turboRFP was fused with an *act-3* promoter region (4.5 kb from V:11073234 to 11077791). The germ line of wild-type animals was injected with the generated plasmids (10 ng ul$^{-1}$). The transgenic lines were subsequently crossed with *act-5(ptc)* mutants to transfer the extrachromosomal array to the mutant background.

## Confocal microscopy
A Zeiss LSM 700 confocal microscope was used to image adult worms.

## RNA interference mediated knockdown
RNAi was performed by feeding double-stranded RNA-expressing bacteria at 25°C from the early larval stage through adulthood (60–75 hr) as previously described (*Fraser et al., 2000*). For the genes whose knockdown from an early larval stage caused lethality or sterility, we started the RNAi treatment at later stages (L4, adult). Also, for some clones (mv_R05D11.6, sjj2_R05D11.6,

sjj2_Y39G8C.1), we diluted the double-stranded RNA-expressing bacteria with empty vector (L4440)-containing bacteria, in order to obtain milder effects. RNAi constructs were obtained from available libraries (Source BioScience) and verified by sequencing. RNAi clones used in this study are listed in the key resources table.

## CRISPR/Cas9 induced mutations

To generate the CRISPR/Cas9-induced *unc-89* deletion allele (*bns7000*), two sgRNAs (final concentration 4 uM each) were injected with Cas9 protein (0.35 ug/ul), and a *dpy-10* sgRNA (2.5 uM) was used as a co-injection marker along with a repair oligo (PSdpy-10-PS; 0.73 uM) (*Dickinson and Goldstein, 2016*).

sgRNA1: 5'-GGTAGTTAGCGACCCCATGAGGG-3'.
sgRNA2: 5'-ACAGACTGGTAAACAAACGAGGG-3'

The following primers were used for genotyping: dunc-89–1 forward (5'-ATACCACCACATGTC TCTTC-3'), dunc-89–2 forward (5'-GCTAAAAGTCAGAGTTCCAC-3'), dunc-89–3 reverse (5'- GGA TGGGTTTACATAAAAT-3'), dunc-89–4 reverse (5'-TGAAAAAGAAACAACAAAA-3'), dunc-89–5 forward (5'-TAACAAAAAGCTCAAAATG-3'), dunc-89–6 reverse (5'-GGATAGATTTCTGTTGGAGA-3'). The external primers flank a 19612 bp region in wild types and amplify a 3601 bp fragment in *bns7000* mutants. The internal primers with different combinations amplify 500-2600 bp products in wild types.

## Double mutant analysis

All the double mutants exhibited gene expression levels as in the RNAi treated animals with one exception. *act-5(ptc); nrde-3* double mutants exhibited *act-5* mRNA levels as in the RNAi experiments but also some upregulation of the adapting gene, unlike what was observed in the RNAi experiments (*Figures 4* and *5*, *Figure 5—figure supplement 1A and B*). One possible explanation is related to an alternative start site of *nrde-3* (*Tourasse et al., 2017*) which might be used only in some tissues and thus could lead to some protein function in the allele used in our study.

## Statistical evaluation

To calculate the significance of the differences for the expression data, we performed two-tailed Student's t-test. Mean ± SEM is indicated in graphs. All statistical analyses were implemented in the program Statistica v. 9. Graphs were generated in Prism5.

## Gene structure visualization

The *act-5* and *unc-89* loci were visualized using the GSDS gene structure visualization tool (*Guo et al., 2007*).

## Acknowledgements

We thank Z Jiang and A Rossi for discussion and comments on the project; G Jakutis, A Rossi, T Sztal, and W Stainier for comments on the manuscript; A McQueen and A Artyukin for some of the *act-5* mutant strains; P Goumenaki and B Grohmann for experimental assistance. Some strains were provided by the CGC, which is funded by the NIH Office of Research Infrastructure Programs (P40 OD010440). The *unc-89(bns7000)* deletion allele was generated in the genome engineering facility, Max Planck Institute of Molecular Cell Biology and Genetics, Dresden, Germany. Research in the Stainier laboratory is supported in part by the Max Planck Society and the European Research Council, ERC AdG 694455-ZMOD.

## Additional information

### Competing interests

Didier YR Stainier: Senior editor, *eLife*. The other authors declare that no competing interests exist.

## Funding

| Funder | Grant reference number | Author |
|---|---|---|
| Max-Planck-Gesellschaft | | Didier YR Stainier |
| H2020 European Research Council | 694455 | Didier YR Stainier |

The funders had no role in study design, data collection and interpretation, or the decision to submit the work for publication.

## Author contributions

Vahan Serobyan, Conceptualization, Data curation, Formal analysis, Validation, Visualization, Methodology, Writing - original draft, Writing - review and editing; Zacharias Kontarakis, Conceptualization, Data curation, Formal analysis, Validation, Visualization, Methodology, Writing - review and editing; Mohamed A El-Brolosy, Data curation, Formal analysis, Methodology, Writing - review and editing; Jordan M Welker, Ann M Wehman, Validation, Writing - review and editing; Oleg Tolstenkov, Data curation, Methodology; Amr M Saadeldein, Data curation; Nicholas Retzer, Data curation, Validation; Alexander Gottschalk, Supervision; Didier YR Stainier, Conceptualization, Resources, Supervision, Funding acquisition, Investigation, Writing - original draft, Project administration, Writing - review and editing

## Author ORCIDs

Vahan Serobyan https://orcid.org/0000-0001-5055-8422
Oleg Tolstenkov https://orcid.org/0000-0002-6484-9965
Amr M Saadeldein https://orcid.org/0000-0002-5971-2083
Alexander Gottschalk http://orcid.org/0000-0002-1197-6119
Ann M Wehman http://orcid.org/0000-0001-9826-4132
Didier YR Stainier https://orcid.org/0000-0002-0382-0026

## Decision letter and Author response

Decision letter https://doi.org/10.7554/eLife.50014.sa1
Author response https://doi.org/10.7554/eLife.50014.sa2

# Additional files

## Data availability

All data generated or analyzed during this study are included in the manuscript and supporting files.

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
