## [Decision Letter]

**Acceptance summary:**

The earlier discovery of what the Stainier lab calls transcriptional adaptation helped to solve confusion about discrepancies between the effects of genetic germline mutations and transient knockdowns of genes in vertebrate. This study now shows that this remarkable phenomenon is not an oddity of vertebrates, but is also detectable in invertebrates (here, nematodes). In addition to demonstrating similar genetic requirements as in vertebrates, the authors report the involvement of factors important for small RNA maturation and transport into the nucleus, which provides new mechanistic insight into transcriptional adaptation.

**Decision letter after peer review:**

Thank you for submitting your article "Transcriptional adaptation in *Caenorhabditis elegans*" for consideration by *eLife*. Your article has been reviewed by 2 peer reviewers, and the evaluation has been overseen by a Reviewing Editor and Jessica Tyler as the Senior Editor. The following individuals involved in review of your submission have agreed to reveal their identity: Michael Patrascheck (Reviewer #1); Scott Kennedy (Reviewer #2).

The reviewers have discussed the reviews with one another and the Reviewing Editor has drafted this decision to help you prepare a revised submission.

In this manuscript the authors describe the discovery of transcriptional adaptation in *C. elegans*. They show for two independent case that mutations in mRNAs of either *act-5* or *unc-89* that destabilize the mRNA can lead to a compensatory increase in related genes. They then use a limited RNAi screen to identify components related to RNA metabolism to be required in different aspects of transcriptional adaptation.

The paper is generally interesting to read and certainly attractive to a wider audience. There are however some aspects – including some critical control experiments – that need to be addressed.

1) Any particular reasons why the strains used in Figure 2 were not used to validate some of the identified factors in Table 1? Especially the *act-3* expression in the gut is a very clear and powerful phenotype. The normalization necessary for qPCR experiments does not adequately account for global RNA levels changes which can be reasonably expected because many of the factors investigated are related to RNA stability. An independent validation seems necessary for at least some factors. The gut-miss-expression phenotype of *act-3* seems ideal to validate some of the RNAi clones that prevent transcriptional adaptation.

2) The authors use RNAi to implicate the NMD and nuclear RNAi RNA surveillance pathways in transcriptional activation in *C. elegans*. Publicly available mutant alleles need to be used to confirm a role for these pathways in transcriptional activation. The worry is that RNAi knockdown induces partial knockdowns, is differentially effective in different tissues, can trigger off-targets silencing, and might trigger indirect and pleiotropic defects due to titration of the RNAi machinery. For example it is noted that smg-2/upf1 knockdown didn't suppress NMD of the *unc-89 ptc* allele. SMG-2/UPF1 is an obligatory component of the NMD machinery and, therefore, a failure to block NMD after SMG-2/Upf1 RNAi is likely a consequence of partial kd- and not differential requirements for SMG-2/Upf1. This concern is fairly easy to remedy. The authors should ask if animals harboring mutations in NMD and nuclear RNAi factors fail to undergo transcriptional adaptation. qRT-PCR to measure RNA levels of *act-3* or *sax-3* +/- NMD/nuclear RNAi null mutations would be proper. Monitoring *act-5::gfp* and *act-3::gfp* +/- NMD/nuclear RNAi would also be proper. Finally, the authors state that *act-5(ptc)* and *unc-89(ptc)* animals do not show their expected loss-of-function phenotypes… presumably because of transcriptional upregulation of homologous genes. Therefore, loss of the adaptation machinery (NMD/nuclear RNAi) in *act-5(ptc)* and *unc-89(ptc)* animals would be expected to uncover the *act-5(ptc)* and *unc-89(ptc)* lof phenotypes.

3) The data shown in Figure 4 may not be sufficient to support the author's claims that small RNAs drive transcriptional adaptation in mammals. The paper does not absolutely need this data in order to be published. Hence, one obvious option is to just remove the data from the paper. Another option would be to soften conclusions drawn from the data. Should the data stay as is, the authors should clarify the model further in the text as well as provide additional experimental support for the model. Clarification; the authors claim that the various AGO/Dicer knockdowns do not affect expression of Rela, but the data say otherwise. All four kds appear to change Rela expression and 2-3 of these changes are statistically significant (Figure 4B). Experimental support; given how AGO-1/2/3 are thought to function, I don't understand why kd of each of the three AGOs should block transcriptional adaptation. Is the idea that all AGOs act sequentially in a linear pathway to direct adaptation? Because of this perplexing result, the authors would need to provide additional data supporting the idea that mammalian AGOs drive adaptation. The authors could show that AGO depletion using an independent approach (maybe auxin/degron or Ago1/3 knockout MEFs) also blocks transcriptional adaptation. Probably easier to just edit text or remove the data.

There are three additional items that it would be nice for the authors to address, but which are not absolutely necessary for publication. These experiments are mentioned because they are all fairly straightforward and their inclusion would make the paper much stronger.

1) It would be fairly simple to provide data suggesting that up-regulation of homologous genes during "transcriptional activation" is actually due to changes in transcription by using the RNA samples (already on hand) to conduct qRT-PCR looking at *act-3/unc-89* pre-mRNA levels. If the effect is transcriptional, intron containing nascent RNAs should change predictably.

2) If the model is correct, then changes in gene expression seen in animals harboring *act-5/unc-89 ptc* alleles should be extremely specific; i.e. limited to just homologs (because up-regulation of homologs rescues mutant phenotypes- so pleiotropic changes in gene expression should be quite minimal). An RNA-seq experiment would be an elegant way to show that this is indeed the case.

3) CRISPR/Cas9-based gfp tagging of genes is a better and more accurate way to readout gene expression than repetitive arrays integrated into random sites in the genome. Figure 2 would be more convincing if an *act-3::gfp* CRISPR line were used in lieu of a transgene. Or, RNA FISH showing upregulation of the *act-3* mRNA in the intestine in *act-5 ptc* animals would also be more convincing than the current data.

---

## [Author Response]

The paper is generally interesting to read and certainly attractive to a wider audience. There are however some aspects – including some critical control experiments – that need to be addressed.1) Any particular reasons why the strains used in Figure 2 were not used to validate some of the identified factors in Table 1? Especially the act-3 expression in the gut is a very clear and powerful phenotype. The normalization necessary for qPCR experiments does not adequately account for global RNA levels changes which can be reasonably expected because many of the factors investigated are related to RNA stability. An independent validation seems necessary for at least some factors. The gut-miss-expression phenotype of act-3 seems ideal to validate some of the RNAi clones that prevent transcriptional adaptation.

Indeed, the expression of the *act-3p::rfp* extrachromosomal reporter in the gut of *act-5(ptc)* mutants is a strong phenotype and appears to be an attractive tool that could be used to confirm the factors identified in the RNAi screen. However, we observed transgenerational expression of this extrachromosomal reporter. When we outcrossed *act-5(ptc); Ex[act-3p::rfp]* animals to transfer the extrachromosomal reporter to a wild-type background, we observed that the WT; *Ex[act-3p::rfp]* animals still expressed RFP in their gut for several generations. Similarly, when we tested some of the RNAi clones with *act-5(ptc); Ex[act-3p::rfp]* animals, most worms still exhibited RFP expression in their gut, although some at a clearly reduced level. Thus, the results were not conclusive. In summary, although the *act-5(ptc); Ex[act-3p::rfp]* animals are useful to illustrate transcriptional adaptation, they cannot be used in a typical screen for genes regulating this process (RNAi or genetic).

The transgenerational inheritance phenomenon observed here, which has been previously described in *C. elegans,* requires further investigation in the context of transcriptional adaptation.

2) The authors use RNAi to implicate the NMD and nuclear RNAi RNA surveillance pathways in transcriptional activation in *C. elegans*. Publicly available mutant alleles need to be used to confirm a role for these pathways in transcriptional activation. The worry is that RNAi knockdown induces partial knockdowns, is differentially effective in different tissues, can trigger off-targets silencing, and might trigger indirect and pleiotropic defects due to titration of the RNAi machinery. For example it is noted that smg-2/upf1 knockdown didn't suppress NMD of the unc-89 ptc allele. SMG-2/UPF1 is an obligatory component of the NMD machinery and, therefore, a failure to block NMD after SMG-2/Upf1 RNAi is likely a consequence of partial kd- and not differential requirements for SMG-2/Upf1. This concern is fairly easy to remedy. The authors should ask if animals harboring mutations in NMD and nuclear RNAi factors fail to undergo transcriptional adaptation. qRT-PCR to measure RNA levels of act-3 or sax-3 +/- NMD/nuclear RNAi null mutations would be proper. Monitoring act-5::gfp and act-3::gfp +/- NMD/nuclear RNAi would also be proper. Finally, the authors state that act-5(ptc) and unc-89(ptc) animals do not show their expected loss-of-function phenotypes… presumably because of transcriptional upregulation of homologous genes. Therefore, loss of the adaptation machinery (NMD/nuclear RNAi) in act-5(ptc) and unc-89(ptc) animals would be expected to uncover the act-5(ptc) and unc-89(ptc) lof phenotypes.

We fully agree with the reviewers on these points and had been generating double mutant data to incorporate into the paper. We focused on mRNA decay and sRNA biogenesis genes and the data are shown in Table 2 and Table 2—source data 1. In general, we observed effects similar to those using RNAi. In addition, we observed that double mutants that failed to undergo transcriptional adaptation phenocopied the *act-5* and *unc-89* deletion alleles (i.e., those that fail to upregulate the adapting gene), further illustrating the physiological relevance of transcriptional adaptation.

3) The data shown in Figure 4 may not be sufficient to support the author's claims that small RNAs drive transcriptional adaptation in mammals. The paper does not absolutely need this data in order to be published. Hence, one obvious option is to just remove the data from the paper. Another option would be to soften conclusions drawn from the data. Should the data stay as is, the authors should clarify the model further in the text as well as provide additional experimental support for the model. Clarification; the authors claim that the various AGO/Dicer knockdowns do not affect expression of Rela, but the data say otherwise. All four kds appear to change Rela expression and 2-3 of these changes are statistically significant (Figure 4B). Experimental support; given how AGO-1/2/3 are thought to function, I don't understand why kd of each of the three AGOs should block transcriptional adaptation. Is the idea that all AGOs act sequentially in a linear pathway to direct adaptation? Because of this perplexing result, the authors would need to provide additional data supporting the idea that mammalian AGOs drive adaptation. The authors could show that AGO depletion using an independent approach (maybe auxin/degron or Ago1/3 knockout MEFs) also blocks transcriptional adaptation. Probably easier to just edit text or remove the data.

We agree with the reviewers that more work needs to be done to confirm the data shown in the original Figure 4; for this reason, we have now removed these data from the manuscript.

There are three additional items that it would be nice for the authors to address, but which are not absolutely necessary for publication. These experiments are mentioned because they are all fairly straightforward and their inclusion would make the paper much stronger.1) It would be fairly simple to provide data suggesting that up-regulation of homologous genes during "transcriptional activation" is actually due to changes in transcription by using the RNA samples (already on hand) to conduct qRT-PCR looking at act-3/unc-89 pre-mRNA levels. If the effect is transcriptional, intron containing nascent RNAs should change predictably.

We have now performed qPCR experiments and found higher pre-mRNA levels of the adapting genes in *act-5(ptc)* and *unc-89(ptc)* mutants compared to WT, suggesting that increased transcription is causing the increase in mRNA levels.

2) If the model is correct, then changes in gene expression seen in animals harboring act-5/unc-89 ptc alleles should be extremely specific; i.e. limited to just homologs (because up-regulation of homologs rescues mutant phenotypes- so pleiotropic changes in gene expression should be quite minimal). An RNA-seq experiment would be an elegant way to show that this is indeed the case.

In our published transcriptome analyses of wild type versus mutant mouse cell lines (El-Brolosy et al., 2019), we observed the differential expression of hundreds of genes. As discussed in the paper, these gene expression changes can be grouped into at least three categories: (1) those that are a direct or indirect result of protein function loss, or forms of compensation other than transcriptional adaptation, (2) those that are a result of transcriptional adaptation, and (3) those that are an indirect result of transcriptional adaptation. Thus, it is unlikely that the adapting genes we analyzed (*act-3* and *sax-3*) are the only genes whose expression is changed in *act-5(ptc)* and *unc-89(ptc)* mutants, respectively. To test specificity for this study, we measured *unc-89* and *sax-3* mRNA levels in *act-5(ptc)* mutants, as well as *act-5* and *act-3* expression levels in *unc-89(ptc)* mutants. We did not observe any significant difference (Figure 3—figure supplement 3), indicating that the expression changes are not genome wide.

3) CRISPR/Cas9-based gfp tagging of genes is a better and more accurate way to readout gene expression than repetitive arrays integrated into random sites in the genome. Figure 2 would be more convincing if an act-3::gfp CRISPR line were used in lieu of a transgene. Or, RNA FISH showing upregulation of the act-3 mRNA in the intestine in act-5 ptc animals would also be more convincing than the current data.

We obtained several GFP-tagged *act-3* lines from Craig Mello, but so far have not observed GFP expression in the gut when crossed into the *act-5(ptc*) mutant background. This discrepancy could be due to differences in expression levels of the transgene vs. the endogenous gene (whole animal qPCR data indicate that the extrachromosomal transgene is upregulated 256-fold vs. 1.5-fold for the endogenous gene in the *act-5(ptc)* background). Given the relatively high autofluorescence of the intestine, a relatively minor increase of the endogenous gene in the GFP-tagged *act-3* lines may not be visible over background. It is also possible that the insertion of GFP in the *act-3* locus interferes with the transcriptional adaptation response. Thus, further work is required to obtain the optimal reagent to pursue this question.

We agree that RNA FISH would be a better approach. However, the high level of sequence similarity of *act-3* with other actin genes makes it challenging to design enough *act-3*-specific probes required for the experiment.